# Antimicrobial Resistance of *Escherichia coli* and *Pseudomonas aeruginosa* from Companion Birds

**DOI:** 10.3390/antibiotics9110780

**Published:** 2020-11-06

**Authors:** Lorena Varriale, Ludovico Dipineto, Tamara Pasqualina Russo, Luca Borrelli, Violante Romano, Stefano D’Orazio, Antonino Pace, Lucia Francesca Menna, Alessandro Fioretti, Antonio Santaniello

**Affiliations:** 1Department of Veterinary Medicine and Animal Productions, Federico II University of Naples, 80134 Naples, Italy; lorena.varriale@unina.it (L.V.); ludovico.dipineto@unina.it (L.D.); russo.tamara@gmail.com (T.P.R.); luca.borrelli@unina.it (L.B.); antonino.pace@unina.it (A.P.); menna@unina.it (L.F.M.); fioretti@unina.it (A.F.); 2Freelancers of the AvianVet, Medicina Aviare Association, 84085 Mercato San Severino, Italy; romano.violante@gmail.com (V.R.); stefano.dorazio@outlook.it (S.D.)

**Keywords:** birds, antimicrobial resistance, bacteria, risk, public health, One Health

## Abstract

Antimicrobial resistance is a public health concern worldwide and it is largely attributed to the horizontal exchange of transferable genetic elements such as plasmids carrying integrons. Several studies have been conducted on livestock showing a correlation between the systemic use of antibiotics and the onset of resistant bacterial strains. In contrast, although companion birds are historically considered as an important reservoir for human health threats, little information on the antimicrobial resistance in these species is available in the literature. Therefore, this study was aimed at evaluating the antimicrobial resistance of *Escherichia coli* and *Pseudomonas*
*aeruginosa* isolated from 755 companion birds. Cloacal samples were processed for *E. coli* and *P. aeruginosa* isolation and then all isolates were submitted to antimicrobial susceptibility testing. *P. aeruginosa* was isolated in 59/755 (7.8%) samples, whereas *E. coli* was isolated in 231/755 (30.7%) samples. Most strains showed multidrug resistance. This study highlights that companion birds may act as substantial reservoirs carrying antimicrobial resistance genes which could transfer directly or indirectly to humans and animals, and from a One Health perspective this risk should not be underestimated.

## 1. Introduction

Antimicrobial resistance (AMR) is a serious concern compromising the empirical treatment of infections and resulting in a lack of effective antibiotics and high medical expenses [1].

The rapid emergence of resistant bacteria is occurring worldwide, endangering the efficacy of antibiotics, which have transformed medicine and saved millions of lives. Many decades after the first patients were treated with antibiotics, bacterial infections have again become a threat. The antibiotic resistance crisis has been attributed to the overuse and misuse of these medications, as well as a lack of new drug development by the pharmaceutical industry due to reduced economic incentives and challenging regulatory requirements [2,3]. The US Center for Disease Control and Prevention (CDC), the European Centre for Disease Prevention and Control (ECDC) and the World Health Organization (WHO) are classifying infections caused by multidrug-resistant (MDR) bacteria as an alarming global disease and a worldwide public health problem [4,5,6,7]. Because of the abuse and misuse of antibiotics both in humans and livestock, and the consequent release in the environment, selected microorganisms have acquired resistance over time by mutation or horizontal transfer of mobile genetic elements carrying resistance genes. The detection of specific mechanisms of resistance by molecular methods has become crucial, both from an epidemiological and a therapeutical point of view. The discovery of novel molecules as an alternative strategy to antibiotics represents a high priority, as well as other important measures of surveillance and control [8,9]. Coordinated efforts to implement new policies, renew research efforts, and pursue steps to manage the crisis are urgently needed [9].

Several studies have been conducted on livestock showing the correlation between the systemic use of antibiotics and the onset of resistant bacterial strains [6]. In contrast, although companion birds are historically considered an excellent animal reservoir, poor information on the antimicrobial resistance in these species is available in the literature [10]. This topic has been largely investigated in wild birds, specifically in birds of prey, waterfowl and passerines, which seem to be a remarkable reservoir of multiresistant *E. coli* strains, representing a notable risk for human and animal health by the spread of these bacteria to waterways and other environmental sources via their fecal deposits [11,12].

In Italy, according to data from the Assalco−Zoomark 2017 report [13], pet birds are about thirteen million, almost twice the number of dogs and cats. In pet bird farms, the administration of antibiotics without medical control is a very common practice, contributing to the increasing resistance rates. In the light of the above considerations, this study was aimed at evaluating antimicrobial resistance of *Escherichia coli* and *Pseudomonas aeruginosa* isolated from companion birds to better understand the epidemiological role of these species in the spread of multidrug resistant bacteria between animals, humans and the environment.

## 2. Results

A total of 59/755 (7.8%, 95% confidence interval = 6.05–10.02%) samples were positive for *P. aeruginosa*, whereas a total of 231/755 (30.6%, 95% CI = 27.35–34.04%) samples were positive for *E. coli*. Additionally, a few strains of Gram-negative bacteria such as *Pantoea* spp., *Serratia* spp, *Morganella* spp., and *Citrobacter* spp. were occasionally isolated.

With respect to *P. aeruginosa*, 45/59 (76.3%) strains were resistant to amoxycicillin/clavulanic acid (AMC; 30 μg) and to sulfamethoxazole-trimethoprim (SXT; 25 μg), 42/59 (71.2%) were resistant to doxycycline (DO; 30 μg), 46/59 (78%) were resistant to enrofloxacin (ENR; 5 μg), 17/59 (28.9%) were resistant to gentamicin (CN; 10 μg) and 48/59 (81.3%) were resistant to oxytetracicline (OT; 30 μg). The majority of strains showed multidrug resistance.

Among the *E. coli* isolates, 118/231 (51.1%) were resistant to amoxycicillin/clavulanic acid (AMC; 30 μg), 127/231 (55%) were resistant to sulfamethoxazole-trimethoprim (SXT; 25 μg), 132/231 (57.1%) were resistant to doxycycline (DO; 30 μg), 92/231 (40%) were resistant to enrofloxacin (ENR; 5 μg), 61/231 (26.4%) were resistant to gentamicin (CN; 10 μg), 147/231 (63.6%) were resistant to oxytetracicline (OT; 30 μg). For *E. coli*, most strains were also multidrug resistant. Results are summarized in Table 1 and Table 2.

## 3. Discussion

In this study *E. coli* and *P. aeruginosa* were the most frequently isolated bacteria from companion birds with a prevalence of 30.7% and 7.8%, respectively. These microorganisms are the most abundant facultative bacterial species in the normal microbiota of the large intestine of animals and humans [14,15,16]. Particularly, *E. coli* is one of the most pathogenic bacterial species in cage birds causing aerosacculitis, polyserositis, septicemia and other mainly extraintestinal diseases [17], whereas *P. aeruginosa* is ubiquitous in aviaries and, under favorable conditions, acts as an opportunistic pathogen. It may occur with localised infections such as rhinitis, sinusitis and laryngitis or can be associated with septicemia and hemorrhagic enteritis. Enrofloxacin, gentamicin, ceftazidime, amikacin and piperacillin are the most common antimicrobials used as treatment [18]. Antibiotic resistance of Gram-negative bacteria has been largely reported in avian medicine, especially in poultry, but data available in companion birds are very scant. Our findings are consistent with those reported by Sigirci et al. [19], who isolated *E. coli* from 37.7% of the examined companion birds, with the majority of the isolates resistant to tetracycline (84%) followed by sulfamethoxazole/trimethoprim (46%), streptomycin (34%), and kanamycin (25%). Furthermore, Di Francesco et al. [20], evaluated the AMR of Gram-negative species isolated from 456 domestic canaries, showing multiple resistance, especially against amoxycillin, erythromycin, spiramycin, tiamulin, and tylosin [20]. Our results are in line with this study regarding the multidrug resistance exhibited by *E. coli* strains isolated from canaries, but not for the antimicrobials, which were amoxycillin/clavulanic, sulfamethoxazole-trimethoprim, doxycycline, enrofloxacin, gentamicin and oxytetracycline.

*E. coli* is usually isolated in the gut of healthy pigeons and in many studies is considered as an antibacterial resistance indicator. Ghanbarpour et al. [21] characterized the AMR genotypes in relation with phenotypic traits of *E. coli* strains isolated from household pigeons. Approximately half of the isolates were resistant to three or more antibiotics (40.1%), in particular to tetracycline (98%), cefotaxime (49.3%), kanamycin (34.2%), trimethoprim-sulphamethoxazole (28.2%), enrofloxacin (17.1%), gentamicin (11.1%) and florfenicol (7.8%) [21]. According to these findings, our study showed a high prevalence of MDR *E. coli* strains, with the highest rates for oxytetracycline (70.4%) and trimethoprim-sulfamethoxazole (61.7%). Oxytetracycline, florfenicol and trimethoprim-sulfamethoxazole are the most common molecules administered to pigeons in the treatment of respiratory, digestive and pyogenic infections, contributing to the occurrence of MDR strains, in addition to other environmental sources [22]. Domestic pigeons are likely to harbor resistance genes which could transfer directly or indirectly to humans and animals, and from a One Health perspective this risk should not be underestimated.

A study conducted by Carroll et al. [23], assessed the antimicrobial resistance of *Escherichia coli* in wild birds. Among the examined species, starlings showed the highest prevalence of AMR (5.4%), with tetracycline and streptomycin as predominant resistant phenotypes. Even if molecular processing highlighted that all isolates belonged to phylogenetic group B1, this commensal *E. coli* group is often responsible for both intestinal and extraintestinal infections in different animal species. Commensal *E. coli* strains may harbor antimicrobial resistance determinants, act as reservoirs of resistance, and at a later stage transfer these resistance features to pathogenic bacteria. Even if wild birds are not exposed to antibiotics directly, contact with sewage or animal manure could explain the acquisition of resistance genes and especially migratory birds can contribute to their pandemic spread. Pandemic ESBL-producing *E. coli* clones or plasmids shared by humans, domestic animals, and wildlife further strengthen this hypothesis [24]. With respect to *E. coli* strains isolated in our study from birds of prey (75%), our results are not completely in accordance with those of Giacopello et al. [25], who isolated *E. coli* (53.1%) in raptors admitted to a wildlife rescue center and reported trimethoprim/sulfamethoxazole as the most frequent resistance displayed by Enterobacteriaceae (83.7%) [25], whereas we mentioned amoxycillin/clavulanic acid (33%) as the highest percentage of AMR.

MDR phenotypes are frequently encountered in *P. aeruginosa* causing nosocomial infections. This opportunistic pathogen is considered as “critical” and the identification of new antibiotics is essential to overcome its MDR properties [26]. To our knowledge, data on the AMR profiles of this microorganism in companion birds are not available, even if the possibility to transmit infections and resistant traits to other species, humans included, is a public health concern that should require more attention. Our results show that 7.8% of the examined birds were positive for *P. aeruginosa*, with all the strains resistant to at least one antibiotic and the majority showing multidrug resistance with rates up to 100%. In a study conducted by Vidal et al. [27], *P. aeruginosa* (7%) was detected in birds of prey with systemic infection and oral lesions, whereas in our study the prevalence was higher (25%), even if the raptors appeared clinically healthy. Most strains displayed resistance to all the antimicrobials tested, except one to gentamicin. On the contrary, the above-mentioned study reported 100% resistance to clindamycin and 21% to gentamicin [27].

### Limits

The study conducted, while showing a novel feature for the avian species included, has some limits namely, a large difference in the number of sampled avian species in order to speculate the results within different families or species and a lack of molecular characterization of resistance genes.

## 4. Materials and Methods

### 4.1. Sampling

During the period January 2016–December 2018, cloacal swabs were collected from a total of 735 clinically healthy birds. Sampled animals belonged to the following families and species: Fringillidae (*Carduelis carduelis, Serinus canaria*), Estrildidae (*Erythrura gouldiae*, *Lonchura striata domestica, Taeniopygia guttata*), Psittacidae (*Melopsittacus undulatus*, *Agapornis roseicollis*) and Columbidae (*Columba livia domestica*). All the birds analyzed were selected from different farms located in Campania region (southern Italy), separated by species and kept in cages. As stated by their respective owners, no birds were on antibiotic treatment at the time of sampling and had not received antibiotic treatments in the previous three months. Samples were collected by sterile cotton-tipped swabs. Each sample was stored in Amies Charcoal Transport Medium (Oxoid, Basingstoke, United Kingdom) at 4 °C, transported to the laboratory, and analyzed within 2 h of collection. Furthermore, twenty birds of prey (*Buteo buteo, Accipiter gentilis, Falco peregrinus*) housed for falconry were also sampled as described previously.

### 4.2. Isolation and Identification of Bacteria

Cloacal samples were inoculated into Buffered-Pepton Water (BPW, Oxoid, Milan) and incubated at 37 °C for 24 h. Cultures obtained were plated onto MacConkey agar and Cetrimide agar (Oxoid, Milan), and incubated at 37 °C for 24 h. Suspected *E. coli* colonies were streaked onto Tryptone Bile X-Glucuronide and incubated at 42 °C for 24 h. Finally, all isolates were biochemically identified by using API 20E system (BioMèrieux, Marcy l’Etoile, France), whereas potential *Pseudomonas* spp. colonies were submitted to oxidase test and processed by biochemical identification by API 20 NE system (BioMèrieux, Marcy l’Etoile, France). In addition, all laboratory procedures were carried out according to UNI EN ISO 9001:2015 (Cert. N. 317jSGQ10)

### 4.3. Antimicrobial Susceptibility Tests

All isolates were submitted to antimicrobial susceptibility testing using the disc diffusion method according to Clinical Laboratory Standards Institute (CLSI 2012) [28]. The antimicrobials tested were amoxycillin/clavulanic acid (AMC; 30 μg), sulfamethoxazole-trimethoprim (SXT; 25 μg), doxycycline (DO; 30 μg), enrofloxacin (ENR; 5 μg), gentamicin (CN; 10 μg), and oxytetracycline (OT; 30 μg). *E. coli* ATCC 25922 and *P. aeruginsa* ATCC15442 were used as control strains in each experiment. The inhibition zones were measured and scored as sensitive, intermediate susceptibility and resistant according to the CLSI documents (CLSI, 2014) [29].

## 5. Conclusions

Based on our findings, we highlight the role of companion birds as potential vectors and reservoirs in the spread and transmission of AMR and emphasize the importance of surveillance programs to prevent the impact of this threat on public health [30]. We would also encourage further, deeper studies in this field to be conducted in order to characterize the plasmid profiles eventually associated with the multidrug resistant phenotypes. In fact, identifying the specific mechanisms that underlie antimicrobial resistance in these species would provide additional information about their role in the animal−human−ecosystem interface. 

In our opinion, a One Health approach is increasingly needed, which involves collaboration between veterinarians, doctors, public health professionals and epidemiologists, and may represent a more effective and shared opportunity for the global challenge to AMR that currently affects humans, animals and the ecosystem in a transversal way.

## Figures and Tables

**Table 1 antibiotics-09-00780-t001:** Prevalence of *P. aeruginosa* strains and percentage of AMR phenotypes in the examined animals.

Examined Animals (number)	Acronym for Antibiotics Number of Positive Samples (%)					
	AMC30 ^1^	SXT25 ^2^	DO30 ^3^	ENR5 ^4^	CN10 ^5^	OT30 ^6^
Fringillidae (388)	38/45 (84.4)	37/45 (82.2)	37/45 (82.2)	40/45 (88.8)	14/45 (31.1)	39/45 (86.6)
Estrildidae (52)	4/5 (80.0)	4/5 (80.3)	1/5 (20.0)	1/5 (20.0)	0/5 (0.0)	4/5 (80.0)
Psittacidae (77)	0/3 (0.0)	1/3 (33.3)	1/3 (33.3)	2/3 (66.7)	2/3 (66.7)	2/3 (66.7)
Columbidae (218)	1/1 (100)	1/1 (100)	1/1 (100)	1/1 (100)	1/1 (100)	1/1 (100)
Birds of prey (20)	2/5 (40.0)	2/5 (40.0)	2/5 (40.0)	2/5 (40.0)	0/5 (0.0)	2/5 (40.0)

^1^ Amoxycillin/Clavulanic Acid; ^2^ Trimethoprim/Sulfamethoxazole; ^3^ Doxycycline; ^4^ Enrofloxacin; ^5^ Gentamicin; ^6^ Oxytetracycline.

**Table 2 antibiotics-09-00780-t002:** Prevalence of *E. coli* strains and percentage of AMR phenotypes in the examined animals.

Examined Animals (number)	Acronym for Antibiotics Number of Positive Samples (%)					
	AMC30 ^1^	SXT25 ^2^	DO30 ^3^	ENR5 ^4^	CN10 ^5^	OT30 ^6^
Fringillidae (388)	24/33 (72.7)	13/33 (39.4)	15/33 (45.4)	18/33 (54.5)	12/33 (36.4)	18/33 (54.5)
Estrildidae (52)	5/13 (38.5)	4/13 (30.8)	4/13 (30.8)	4/13 (30.8)	2/13 (15.4)	6/13 (46,1)
Psittacidae (77)	7/8 (87.5)	8/8 (100)	7/8 (87.5)	8/8 (100)	3/8 (37.5)	7/8 (87.5)
Columbidae (218)	77/162 (47.5)	100/162 (61.7)	106/162 (65.4)	62/162 (38.3)	42/162 (25.9)	114/162 (70.4)
Birds of prey (20)	5/20 (33.3)	2/20 (13.3)	0/20 (0.0)	0/20 (0.0)	2/20 (13.3)	2/20 (13.3)

^1^ Amoxycillin/Clavulanic Acid; ^2^ Trimethoprim/Sulfamethoxazole; ^3^ Doxycycline; ^4^ Enrofloxacin; ^5^ Gentamicin; ^6^ Oxytetracycline.

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
