# Peer review of "Antimicrobial Resistance of Escherichia coli and Pseudomonas aeruginosa from Companion Birds"

_antibiotics, 2020, doi:10.3390/antibiotics9110780_

Round 1

Reviewer 1 Report

The manuscript “antibiotics-984512” authored by Varriale et al. examines the patterns of antimicrobial resistance of Escherichia coli and Pseudomonas aeruginosa isolated from companion birds in an attempt to shed some light on the epidemiological role of these species in the spread of multidrug resistant bacteria between animals, humans and environment. I have some comments that could help to improve the manuscript.

1. The authors should indicate which pathogens other than Escherichia coli and Pseudomonas aeruginosa were found in the samples.

2. When writing about any microorganism in a scientific document, the genus name should be written in full upon its first use. For subsequent uses, the genus should be abbreviated to its first letter followed by a period.

3. Scientific names of microorganisms must be written in italics. Please, address this issue in text and tables accordingly.

4. There are some spelling and grammatical errors in the manuscript that could be corrected. For example:

a) P1, L19: Please, change “are” to “is”

b) P3, L93 and L95: Please, change “gram” to “Gram”.

Author Response

Manuscript ID: antibiotics-984512
Manuscript Title: Antimicrobial-Resistance of Escherichia coli and Pseudomonas aeruginosa from Companion Birds

REPLY TO REVIEWER COMMENTS (The authors’ response is reported in red)
Dear Reviewer #1#,
Thank you for your consideration of our manuscript for publication in Antibiotics. We have now revised it in light of your useful comments and suggestions. The following point-by-point response letter- clearly indicating marked in red how and where the changes were made in the manuscript - has been prepared to address your comments. In the Manuscript, all revisions were clearly highlighted using the "Track Changes" function in Microsoft Word.
We look forward to your further disposition.

The manuscript “antibiotics-984512” authored by Varriale et al. examines the patterns of antimicrobial resistance of Escherichia coli and Pseudomonas aeruginosa isolated from companion birds in an attempt to shed some light on the epidemiological role of these species in the spread of multidrug resistant bacteria between animals, humans and environment. I have some comments that could help to improve the manuscript.

Response: We would like to thank the Reviewer for taking the time to review our article and for his approval. We are sure that thanks to your precious suggestions we will be able to improve our article, making it clearer and more fluent for readers.

1.The authors should indicate which pathogens other than Escherichia coli and Pseudomonas aeruginosa were found in the samples.

Response: Thank you for your suggestion. We have added in the text at line 68-70 the following sentence: “Additionally, a few strains of Gram-negative bacteria such as Pantoea spp., Serratia spp, Morganella spp., and Citrobacter spp. have occasionally been isolated.”

2. When writing about any microorganism in a scientific document, the genus name should be written in full upon its first use. For subsequent uses, the genus should be abbreviated to its first letter followed by a period.

Response: We apologize for the lack. We have corrected the inaccuracies accordingly.

3. Scientific names of microorganisms must be written in italics. Please, address this issue in text and tables accordingly.

Response: Scientific names of microorganisms have been corrected as suggested, thanks.

4. There are some spelling and grammatical errors in the manuscript that could be corrected. For example:
a) P1, L19: Please, change “are” to “is”
Response: The correction was carried out accordingly, thanks.
b) P3, L93 and L95: Please, change “gram” to “Gram”.

Response: The correction was carried out accordingly, thanks.

Reviewer 2 Report

The prevalence of drug-resistant bacteria, caused among others by excessive use of antibiotics, is increasing problem due to possibility of transmission of resistant bacteria or their resistance genes between animals and humans via direct or indirect contact, through food/feed and the environment.

For a long time, focus was mostly on antimicrobial resistance (AMR) monitoring in food-producing animals. Recently, pets have been described as potential vehicle of AMR, however data remain scarce, therefore it was found that it is necessary to take a closer look at the situation in companion animals. Approaching any issue from a One Health perspective necessitates looking at the interactions of people, domestic animals including pets, wildlife, plants and our environment.

The work is interesting mainly because it deals with resistance in companion birds, which together with other companion animals is a little explored area, especially in commensal bacteria such as E. coli considered as a reservoir of mobile genetic elements carrying antibiotic resistance genes and indicator for the surveillance and spread of AMR for pathogenic bacteria.

Despite the importance of the above-mentioned overview of resistance in companion birds, some important data are missing or are un-correctly interpreted:

Firstly, companion birds cannot be considered as sentinel of antimicrobial resistance. The reservoir or vehicle is correct terminology. The wild birds are rather considered as sentinel.

Secondly, there is no information if monitored birds were treated with antibiotics and/or how many percent of the birds examined were treated and what antibiotics or antimicrobial classes were used. In my opinion this is important information, because of bacterial AMR of treated animals may be diametrically different. In the section “Material and Methods” there is only information that “cloacal swabs were collected from clinically healthy birds”.

It is important to positively evaluate the relatively high number of tested strains. On the other hand, as the authors themselves admit, the study conducted has some limits consisting of large difference in the number of sampled avian species in order to speculate the results within different families or species; and lack of molecular characterization of resistance genes.

Important references deals with antimicrobial resistance in companion and/or wild birds are missing and in my opinion it is necessary to complete discussion with mentioned ones:

  1. Diren Sigirci et al./Journal of King Saud University – Science 32 (2020) 1069–1073 https://doi.org/10.1016/j.jksus.2019.09.014

Jing Wang, Zhen-Bao Ma, Zhen-Ling Zeng, Xue-Wen Yang, Ying Huang, Jian-Hua Liu. The role of wildlife (wild birds) in the global transmission of antimicrobial resistance genes. Zoological Research, 2017, 38(2): 55-80. doi: 10.24272/j.issn.2095-8137.2017.003

Lines 91-93:

“Particularly, E. coli is one of the most pathogenic bacterial species in cage birds, whereas P. aeruginosa is ubiquitous in aviaries and, under favorable conditions, acts as an opportunistic pathogen.”

If mentioned bacteria are considered pathogenic, it would be useful to list the most often diseases causing by them and antibiotics used for their treatment.

Line 138-139:

“tested, except one to gentamycin. On the contrary, the above-mentioned study reported 100% of  resistance to clindamycin and 21% to gentamycin”

Correctly is gentamicin.

Author Response

Manuscript ID: antibiotics-984512
Manuscript Title: Antimicrobial-Resistance of Escherichia coli and Pseudomonas aeruginosa from Companion Birds

REPLY TO REVIEWER COMMENTS (The authors’ response is reported in red)
Dear Reviewer #2#,
Thank you for your consideration of our manuscript for publication in Antibiotics. We have now revised it in light of your useful comments and suggestions. The following point-by-point response letter- clearly indicating marked in red how and where the changes were made in the manuscript - has been prepared to address your comments. In the Manuscript, all revisions were clearly highlighted using the "Track Changes" function in Microsoft Word.
We look forward to your further disposition.

The prevalence of drug-resistant bacteria, caused among others by excessive use of antibiotics, is increasing problem due to possibility of transmission of resistant bacteria or their resistance genes between animals and humans via direct or indirect contact, through food/feed and the environment.

For a long time, focus was mostly on antimicrobial resistance (AMR) monitoring in food-producing animals. Recently, pets have been described as potential vehicle of AMR, however data remain scarce, therefore it was found that it is necessary to take a closer look at the situation in companion animals. Approaching any issue from a One Health perspective necessitates looking at the interactions of people, domestic animals including pets, wildlife, plants and our environment.

The work is interesting mainly because it deals with resistance in companion birds, which together with other companion animals is a little explored area, especially in commensal bacteria such as E. coli considered as a reservoir of mobile genetic elements carrying antibiotic resistance genes and indicator for the surveillance and spread of AMR for pathogenic bacteria.

Response: We would like to thank the Reviewer for taking the time to review our article and for his approval. We are sure that thanks to your precious suggestions we will be able to improve our article, making it clearer and more fluent for readers.

Despite the importance of the above-mentioned overview of resistance in companion birds, some important data are missing or are un-correctly interpreted:

Firstly, companion birds cannot be considered as sentinel of antimicrobial resistance. The reservoir or vehicle is correct terminology. The wild birds are rather considered as sentinel.

Response: Thank you for your suggestion. We have replaced the sentinel with a “reservoir” on lines 18 and 52, while we removed it on line 187.

Secondly, there is no information if monitored birds were treated with antibiotics and/or how many percent of the birds examined were treated and what antibiotics or antimicrobial classes were used. In my opinion this is important information, because of bacterial AMR of treated animals may be diametrically different. In the section “Material and Methods” there is only information that “cloacal swabs were collected from clinically healthy birds”.

It is important to positively evaluate the relatively high number of tested strains. On the other hand, as the authors themselves admit, the study conducted has some limits consisting of large difference in the number of sampled avian species in order to speculate the results within different families or species; and lack of molecular characterization of resistance genes.

Response: Thank you, your suggestion allows us to add this important information to the manuscript (lines 162-164).

Important references deals with antimicrobial resistance in companion and/or wild birds are missing and in my opinion it is necessary to complete discussion with mentioned ones:
Diren Sigirci et al./Journal of King Saud University – Science 32 (2020) 1069–1073 https://doi.org/10.1016/j.jksus.2019.09.014

Jing Wang, Zhen-Bao Ma, Zhen-Ling Zeng, Xue-Wen Yang, Ying Huang, Jian-Hua Liu. The role of wildlife (wild birds) in the global transmission of antimicrobial resistance genes. Zoological Research, 2017, 38(2): 55-80. doi:10.24272/j.issn.2095-8137.2017.003

Response: Thank you for pointing out these two interesting references. We added them in the discussion to broaden the comparison with our data, as suggested.

Lines 91-93:
“Particularly, E. coli is one of the most pathogenic bacterial species in cage birds, whereas P. aeruginosa is ubiquitous in aviaries and, under favorable conditions, acts as an opportunistic pathogen.”
If mentioned bacteria are considered pathogenic, it would be useful to list the most often diseases causing by them and antibiotics used for their treatment.

Response: Thank you for your suggestion. We have added the requested information in the text in lines 93-98.

Line 138-139:
“tested, except one to gentamycin. On the contrary, the above-mentioned study reported 100% of resistance to clindamycin and 21% to gentamycin”
Correctly is gentamicin.

Response: we apologize for the inaccuracy; we have corrected the error. Thank you.
